# The Role of Cellular Immunity in the Protective Efficacy of the SARS-CoV-2 Vaccines

**DOI:** 10.3390/vaccines10071103

**Published:** 2022-07-09

**Authors:** Zhongjie Sun, Tingxin Wu, Huangfan Xie, Yuhuan Li, Jinlan Zhang, Xuncheng Su, Hailong Qi

**Affiliations:** 1State Key Laboratory of Elemento-Organic Chemistry, College of Chemistry, Nankai University, Tianjin 300071, China; 1120210470@mail.nankai.edu.cn; 2Newish Technology (Beijing) Co., Ltd., Beijing 100176, China; wutingxin@newishes.com (T.W.); xiehuangfan@newishes.com (H.X.); 3Institute of Medicinal Biotechnology, Chinese Academy of Medical Science and Peking Union Medical College, Beijing 100050, China; liyuhuan@imb.pumc.edu.cn; 4State Key Laboratory of Bioactive Substance and Function of Natural Medicines, Institute of Materia Medica, Chinese Academy of Medical Science and Peking Union Medical College, Beijing 100050, China; zhjl@imm.ac.cn

**Keywords:** SARS-CoV-2, vaccine, cellular immunity

## Abstract

Multiple severe acute respiratory syndrome coronavirus 2 (SARS-CoV-2) vaccines have been approved for clinical use. SARS-CoV-2 neutralizing antibody titers after immunization are widely used as an evaluation indicator, and the roles of cellular immune responses in the protective efficacy of vaccines are rarely mentioned. However, therapeutic monoclonal neutralizing antibodies have shown limited efficacy in improving the outcomes of hospitalized patients with coronavirus disease 2019 (COVID-19), suggesting a passive role of cellular immunity in SARS-CoV-2 vaccines. The synergistic effect of virus-specific humoral and cellular immune responses helps the host to fight against viral infection. In fact, it has been observed that the early appearance of specific T-cell responses is strongly correlated with mild symptoms of COVID-19 patients and that individuals with pre-existing SARS-CoV-2 nonstructural-protein-specific T cells are more resistant to SARS-CoV-2 infection. These findings suggest the important contribution of the cellular immune response to the fight against SARS-CoV-2 infection and severe COVID-19. Nowadays, new SARS-CoV-2 variants that can escape from the neutralization of antibodies are rapidly increasing. However, the epitopes of these variants recognized by T cells are largely preserved. Paying more attention to cellular immune responses may provide new instructions for designing effective vaccines for the prevention of severe disease induced by the break-through infection of new variants and the sequelae caused by virus latency. In this review, we deliberate on the role of cellular immunity against COVID-19 and summarize recent advances in the development of SARS-CoV-2 vaccines and the immune responses induced by vaccines to improve the design of new vaccines and immunization strategies.

## 1. Introduction

In December 2019, a severe infectious pneumonia outbreak occurred in Wuhan, Hubei Province, China, and scientists ultimately determined that the infectious pneumonia was caused by SARS-CoV-2 [1]. On 11 March 2020, the World Health Organization (WHO) officially named the pneumonia caused by SARS-CoV-2 infection as COVID-19 and declared a global pandemic. According to WHO statistics, the number of confirmed COVID-19 patients worldwide has exceeded 300 million, and the number of deaths directly caused by SARS-CoV-2 has exceeded 5 million (https://covid19.who.int/ accessed on 1 March 2022). SARS-CoV-2 has resulted in immeasurable losses for the lives and economy of human society. A variety of therapeutic drugs, such as small-molecule inhibitors and neutralizing antibodies, have been developed for COVID-19 treatment [2,3]. However, due to the high transmission efficiency of SARS-CoV-2 and the continuous emergence of drug-resistant variants, a safe and effective vaccine is still needed to help humans to establish an immune barrier to block the transmission of SARS-CoV-2.

SARS-CoV-2, severe acute respiratory syndrome coronavirus (SARS-CoV), and Middle East respiratory syndrome coronavirus (MERS-CoV), the three viruses that lead to severe respiratory diseases in humans, all belong to the coronavirus family. This family is named after the fact that the spike (S) proteins on the surface of the outer capsular membrane of the virus form a crown-like shape. A sense RNA genome, and nucleoprotein (N), and other nonstructural proteins of the virus are encapsulated within the envelope [4,5]. After entering into the human body, SARS-CoV-2 is endocytosed into host cells mainly through the binding of the S protein to host-cell surface angiotensin-converting enzyme-2 (ACE2) [5,6]. Currently, the S protein is the main target for the design of therapeutic neutralizing antibodies and preventive vaccines, and the effectiveness of targeting the S protein has been confirmed by preclinical and clinical trials [4,6,7,8].

According to WHO statistics, as of November 2021, there were 326 SARS-CoV-2 vaccines under development worldwide (https://www.who.int/publications/m/item/draft-landscape-of-covid-19-candidate-vaccines accessed on 1 November 2021). These SARS-CoV-2 vaccines were developed based on both conventional vaccine platforms, which include inactivated vaccines, subunit vaccines, virus-like particles, and attenuated virus vaccines, and nonconventional vaccine platforms, including messenger RNA (mRNA) vaccines, DNA, and replicable and nonreplicable viral vectors [9]. Currently, vaccines approved for marketing are administered around the world, and most of these vaccines have achieved a preventive efficacy greater than 50% in clinical trials, a benchmark specified by the WHO [10]. However, the preventive efficacy of different vaccines is significantly variable, ranging from 57.5% to 95%. Currently, the studies of approved vaccines mainly use the neutralizing antibody induced by vaccine immunization as an immune evaluation indicator. IgG is the main isotype of antibody that mediates the neutralization of the virus. IgG can be divided into four subtypes (IgG1–4). Different subtypes show different affinities for the activating or inhibiting of Fcγ-receptors, which results in pro-inflammatory responses or in the dampening of inflammatory responses [11]. Specific IgG for SARS-CoV-2 can be detected 7–14 days after the onset of symptoms [12]. For most vaccines, the antibody can reach the peak after the second immunization and then significantly decrease within 4–6 months [13]. Increasing data indicate that cellular immune responses play important roles in the clearance of SARS-CoV-2 and the alleviation of COVID-19 [14]. Here, we summarize the distinction in the induced immune responses and protective efficacy between several vaccines with disclosed clinical trial results.

### 1.1. Humoral and Cellular Immune Responses Induced by SARS-CoV-2 Vaccines

China has developed three inactivated virus vaccines: BBIBP-CorV (Sinovac Biotech), and WIV04 and HB02 (Sinopharm). Randomized phase III clinical data indicated that the protective efficiency of inactivated virus vaccines against disease symptoms after infection was between 51% and 79.4% [15,16,17,18]. The levels of neutralizing antibodies induced by inactivated vaccines after inoculation are similar to the levels of serum neutralizing antibodies in patients infected with SARS-CoV-2, and cellular immune responses targeting SARS-CoV-2 proteins, such as S and N, are induced [19]. Typical nucleic acid vaccines include mRNA-1273, developed by Moderna (Cambridge, MA, USA), and BNT162b2, developed by Pfizer, with clinical inoculation doses of 100 µg and 25 µg, respectively. After inoculation with these mRNA vaccines, high levels of neutralizing antibodies are induced in vaccinated individuals, with titers that are approximately 2–10 times those in convalescent serum from COVID-19 patients. The mRNA vaccine also induces strong type 1 T helper (Th1)-type CD4^+^ T-cell and CD8^+^ cytotoxic T-lymphocyte (CTL) immune responses [7,20,21]. In phase III clinical trials, the protection efficiency of both mRNA-1273 and BNT162b2 was greater than 90% [22]. Several recent comparative studies have reported that after immunization with mRNA-1273, BNT162b2, and Johnson & Johnson’s Ad26.COV2.S adenoviral vector vaccine, the induced neutralizing antibody titers were sequentially decreased, and the induced CTL immune response was sequentially increased. The neutralizing antibody titers induced by the two mRNA vaccines significantly decreased after six months, whereas those of Ad26.COV2.S increased. For the three vaccines, unlike the antibody response, which gradually decreased after vaccination, the cellular immune response was found not to have decreased 8 months after immunization [22,23]. A real-world effectiveness study found that compared with those after BNT162b2 vaccination, the infection rate, symptomatic patient rate, hospitalization rate, severe disease rate, and mortality rate after mRNA-1273 vaccination were reduced by 1.23%, 0.44%, 0.55%, 0.10%, and 0.02%, respectively [24]. A comparative study of the three vaccines based on hospitalization rates reported protective efficacies of mRNA-1273, BNT162b2, and Ad26.COV2.S of 93%, 88%, and 71%, respectively [25]. Another comparative study of inactivated virus vaccines and BNT162b2 indicated that the level of neutralizing antibodies induced by inactivated vaccines was only one-tenth of that induced by the mRNA vaccine and that the cellular immune response induced by inactivated vaccines was about one-half of that induced by the mRNA vaccine; however, the cellular immune responses of the inactivated vaccines were reactive to more SARS-CoV-2 protein epitopes [19,26]. The immune characteristics and protection efficiency of the above vaccines indicate that there are large differences in the induced humoral immunity and cellular immunity among vaccines created using different platforms (Table 1). Compared with inactivated vaccines, nucleic acid vaccines can induce stronger humoral and cellular immune responses. Among various nucleic acid vaccines, adenovirus-delivered DNA vaccines and electroporation-delivered plasmid DNA vaccines can induce strong cellular immune responses, while mRNA vaccines favor humoral immune responses.

Recent studies have found that there is a correlation between a strong antibody immune response and the protection efficiency of the vaccine [27,28,29]. However, two COVID-19 patients diagnosed with X-linked agammaglobulinemia (XLA), which results in no circulating B cells, could also fully recover from SARS-CoV-2 infection [30]. Another observation of SARS-CoV-2-infected patients with multiple sclerosis who used monoclonal antibodies to delete B cells found that these patients had mild–moderate symptoms and could also completely recover [31,32,33]. This body of evidence supports the possibility of controlling SARS-CoV-2 in the absence of neutralizing antibodies. So far, the correlation between the cellular immune response and the protection efficiency of a vaccine remains unclear. The needle-free intradermal injection of a plasmid DNA vaccine developed in India has achieved a protection efficiency greater than 60% in India, where the Delta variant is the dominant strain. This protection efficiency is similar to that reported for mRNA-1273 and BNT162b2. Immune assessments indicated that the level of neutralizing antibodies induced by the DNA vaccine was slightly lower than that in the convalescent serum but that the cellular immune response was stronger, suggesting that the cellular immunity induced by DNA vaccines also plays a role in protecting the host against viral infections [34,35,36]. In a similar vein, although the antibody level of inactivated virus vaccines was significantly lower than that of the mRNA vaccine, a similar effectiveness of inactivated vaccine BBIBP-CorV (Sinopharm) and mRNA vaccine BNT162b2 (Pfizer-BioNTech) against COVID-19-related hospitalizations was observed during the Delta outbreak in the United Arab Emirates, which may have been caused by the greater diversity of cellular immune responses [37]. Next, the possible role of cellular immunity in host defense against SARS-CoV-2 infection is discussed.

**Table 1 vaccines-10-01103-t001:** The immune characteristics and protection efficiency of the different platform-based vaccines (HCS, healthy convalescent serum).

Name of Vaccine	Platform	Spike- or RBD-Specific Antibody Geometric Mean Titer (GMT)	CD4^+^	CD8^+^	Clinical Efficacy
**mRNA-1273**	RNA	1,192,154 compared to 142,140 of HCS	Th1 and Tfh	Potent response	63.0% efficacy for infections (95% CI, from 56.6 to 68.5) and 98.2% (95% CI, from 92.8 to 99.6) efficacy for severe COVID-19 [13,38,39]
**BNT162b2**	RNA	25,006 compared to 602 of HCS	Th1 and Tfh	Potent response	95% efficacy for prevention of COVID-19 (95% CI, from 90.3 to 97.6) [7,13,40]
**BBIBP-CorV**	Inactivated virus	Neutralizing antibody, 228.7; no comparison to that of HCS	CD4^+^ or CD8^+^ response non-distinguishable in humans	CD4^+^ or CD8^+^ response non-distinguishable in humans	83.5% efficacy for infections (95% CI 65.4–92.1) and 100% efficacy for moderate hospitalization [18,19,41]
**INO-4800**	DNA	From 655.5 to 994.2; no comparison to that of HCS	Not significant	Potent response	Not mentioned [42]
**ZyCoV-D**	DNA	884.04; slightly lower than that of HCS	CD4^+^ or CD8^+^ response non-distinguishable in humans	CD4^+^ or CD8^+^ response non-distinguishable in humans	66.6% efficacy for infections (95% CI 47.6–80.7) and 100% efficacy for moderate–severe COVID-19 [36,43]
**Ad5-nCoV (Adenovirus type 5 vector)**	Ad5 (nonreplicating adeno virus)	615.8–1445.8; no comparison to that of HCS	CD4^+^ or CD8^+^ response non-distinguishable in humans	CD4^+^ or CD8^+^ response non-distinguishable in humans	57.5% efficacy (95% CI, from 39.7 to 70) for infections and 91.7% efficacy for severe COVID-19 [44,45]
**Ad26.COV2.S**	Ad26 (nonreplicating adenovirus)	1677–2292 compared to 899 of HCS	Th1	Moderate response	66.1–76% efficacy (95% CI, 75–77%) for infections and 81% efficacy (95% CI, 78–82%) for COVID-19-related hospitalizations [13,46,47,48,49]
**NVX-CoV2373**	Protein subunit	63160 compared to 8344 of HCS	Th1	CD8^+^ response not detected in humans	89.7% efficacy for infections (95% CI, 80.2 to 94.6) 100% efficacy for severe COVID-19 [13,50,51]
**ZF2001**	Protein subunit	2777; no comparison to that of HCS	Th1 and Th2	CD8^+^ response not detected in humans	Not mentioned [52]

### 1.2. Role of Cellular Immunity in the Process of SARS-CoV-2 Infection and COVID-19

To understand the role of cellular immunity in the fight against SARS-CoV-2 infection and COVID-19, it is necessary to understand the cellular immune response process of individuals after SARS-CoV-2 infection. First, SARS-CoV-2 infection is usually accompanied by a decline in absolute CD4^+^ T- and CD8^+^ T-cell counts. A correlation analysis of disease severity, patient prognosis, and T-cell counts revealed that smaller decreases in the CD8^+^ T-cell count led to milder symptoms and better prognosis, suggesting that CD8^+^ T-cell count can be used as a marker of recovery in COVID-19 patients [53,54]. Subsequent studies found that multiple memory-cell subsets targeting different SARS-CoV-2 proteins could be detected in the convalescent blood of COVID-19 patients and that the proportion of virus-specific CD8^+^ T cells in patients with mild disease was higher than that in severe individuals [55,56]. Another group used single-cell sequencing technology to analyze the immune cell subtypes in the bronchoalveolar lavage (BAL) fluid of COVID-19 patients with different disease severities and found clonal CD8^+^ T-cell proliferation in the alveoli of patients with mild disease and a disruption in T-cell subset distribution in severe patients, suggesting that after SARS-CoV-2 infection, the human body produces a memory immune cell response against SARS-CoV-2 and that a virus-specific CD8^+^ T-cell immune response alleviates symptoms rather than aggravating a patient’s condition [57]. Increasing evidence indicates that the cellular immune response plays an important role in protecting the host against SARS-CoV-2 [58]. Study have found that cellular immune responses capable of recognizing the constituent proteins of SARS-CoV-2 exist in individuals who have never been exposed to SARS-CoV-2; however, the exact reason is unclear. Another study has shown that these individuals with pre-existing SARS-CoV-2 cellular immunity have a significantly reduced risk for SARS-CoV-2 infection [59,60]. These data show, to some extent, that the cellular immune response can help individuals to fight against SARS-CoV-2 infection, which is consistent with the results of a previous study, i.e., vaccine-induced airway memory CD4^+^ T cells can protect animals from coronavirus infection [61]. For patients infected with SARS-CoV-2, a correlation analysis of prognosis and T-cell count, and the S-protein-specific T-cell immune response indicated that after SARS-CoV-2 infection, a high average T-cell count and the early appearance of S-specific T cells were positively correlated with a good prognosis and mild symptoms. However, no such correlation between the level of S-specific antibodies and patient symptoms was found in this analysis [62,63]. On the contrary, the lack of SARS-CoV-2-specific CD4^+^ and CD8^+^ T cells was associated with COVID-19 severe disease [64]. These results further underline the role of cellular immunity in controlling the severity of symptoms in patients infected with SARS-CoV-2. In summary, the cellular immune response may play an important role in the prevention of infection and the prevention of severe illness after SARS-CoV-2 infection.

### 1.3. Types and Protective Effect of Cellular Immunity Induced by SARS-CoV-2 Infection and Vaccines

Clearing intracellular viruses requires the lysis of infected cells. CD8^+^ T cells can lyse target cells by inducing apoptosis. Thymus-deficient mice could not completely eliminate influenza virus, and the adoptive transfer of SARS-CoV-specific CD8^+^ T cells to SCID mice could enhance survival and reduce virus titers. Therefore, the removal of virus-infected cells through CD8^+^ T cells is important for the complete remission of infection (Figure 1). In COVID-19 patients, S-, nucleocapsid-, M-, and ORF3a-specific CD8^+^ T cells can be detected in the blood after infection [59,65,66]. These SARS-CoV-2-specific CD8^+^ T cells can be rapidly generated during acute COVID-19, and they can even be detected within one day after infection [67]. Moreover, these specific CD8^+^ T cells show high-intensity cytotoxic functions because they highly express IFN-γ, granzyme B, perforin, and CD107a [67,68]. Similar to COVID-19 patients with SARS-CoV-2 infection, multiple platform vaccines can induce a potent S-specific CD8^+^ T-cell immune response [21,42,69,70]. 

CD4^+^ T cells can differentiate into multiple functional subsets. Different subsets have the functions of promoting B-cell proliferation, antibody subclass conversion, helping CD8^+^ T cells, recruiting innate immune cells, and assisting in tissue repair. In the process of antiviral infection, Th1 and T follicular helper cells (Tfh) subsets are usually differentiated to exert antiviral activity by producing cytokines. Tfh play important roles in the production of neutralizing antibodies and in the instruction of memory-B-cell differentiation for long-term humoral immunity development [71]. SARS-CoV-2-specific effect and memory Tfh cells can be detected in patients with COVID-19. The frequency of Tfh cells is inversely correlated with the severity of COVID-19 [72,73]. In addition to Tfh cells, Th1 CD4^+^ T cells secreting Th1-type cytokines such as IFN-γ, TNF, and IL-2 can also be detected in patients with COVID-19. A study found that patients who died of COVID-19 had fewer IFN-γ-secreting CD4^+^ T cells than those with mild illness and rehabilitation [74]. A variety of different platform vaccines can induce the production of Tfh- and Th1-cell subtypes. After immunization, the vaccine can induce CD8^+^ T cells and a variety of CD4^+^ T-cell subtypes that can weaken the symptoms of COVID-19 at the same time, indicating that the vaccine may play many other positive functions in addition to preventing SARS-CoV-2 infection [7,21,42,69,70].

In conclusion, a variety of identified cell subsets play important roles in limiting virus replication and protecting the host from virus-induced tissue damage. Vaccines developed based on different platforms differ in inducing cell-response subtypes. The possible reasons are related to the types and forms of antigens delivered by different platforms and the type of adjuvant used. For example, after the protein subunit vaccine antigen is phagocyted by APCs, it is mainly degraded in lysosome and presented to CD4^+^ through the MHC II molecule. The CpG adjuvant for this vaccine activates TLR9, which eventually leads to a Th1-type cellular immune response, but it is hard to induce a CD8^+^ cellular immune response (Figure 1) [75].

Furthermore, immunization with CD4^+^ or CD8^+^ single epitope peptide screened from mice induced CD4^+^- or CD8^+^-cell immune responses in mice to protect mice against COVID-19 to a certain degree. These peptides did not cause any detectable humoral immune response. This result further supports the role of cellular immunity in the fight against COVID-19 [76].

Recently, immunization with a self-amplifying RNA vaccine encoding the SARS-CoV-2 N protein that failed to produce S neutralizing antibody responses against the Beta variant protected hamsters from body weight loss and decreased the viral load of hamsters challenged with this VOC [77]. This result indicates that the cellular immune response against the nonstructural protein of SARS-CoV-2 could protect the host from COVID-19.

However, the cellular immune response may also cause tissue damage. For example, adoptive virus-specific CD8^+^ T cells caused lung injury in T- and B-cell-deficient mice due to *rag1* knockout. Therefore, the balance between effective virus clearance and immune-induced injury is very important for the rehabilitation from viral infection. 

### 1.4. SARS-CoV-2 Variants Can Escape Vaccine-Induced Humoral Rather Than Cellular Immunity

The SARS-CoV-2 genome is a single-stranded RNA that is prone to mutations during the replication process. With the prolonged COVID-19 pandemic, the SARS-CoV-2 genome has accumulated numerous mutations, and new variants have emerged. These variants are classified into two types based on transmission ability, pathogenicity, and tolerance to vaccines and drugs, i.e., variants of interest (VOIs) and variants of concern (VOCs). The transmission abilities and pathogenic abilities of VOCs are enhanced, and VOCs have a certain tolerance to therapeutic drugs and vaccines. Currently, five VOCs, namely, Alpha, Beta, Gamma, Delta, and Omicron, have been recorded around the world. Among the mutations accumulated in SARS-CoV-2 VOCs, the primary concern is amino acid mutations in the S protein, because the current vaccines and neutralizing antibodies mainly target the S protein. Mutations in the S protein are likely to reduce the protective efficacy of vaccines and the therapeutic effect of monoclonal antibodies. It was found that the N501Y mutation in the receptor-binding domain (RBD) of the S protein is present in four VOCs, i.e., Alpha, Beta, Gamma, and Omicron. The N501Y mutation enhanced the transmission ability of SARS-CoV-2 by enhancing its affinity to the ACE2 receptor [78]. Moreover, the N501Y mutation also reduced the neutralizing activity of most therapeutic monoclonal antibodies and the serum of COVID-19 convalescents [79,80]. The N439K mutation in the RBD also enhanced the affinity to ACE2 and favored the escape of SARS-CoV-2 from neutralizing antibodies [81]. Another mutation in the RBD, E484K, reduced the neutralizing titers of vaccine recipients and neutralizing titers in the convalescent serum of COVID-19 patients by 2 and 4.5 times, respectively [79,82]. The L452R and Y453F mutations enhanced the fusion of the virus to promote virus reproduction and make the epitope insensitive to cellular immunity [83]. The current VOCs contain one or more of the above mutations. For example, the recently emerged Omicron variant contains many important mutations that promote immune escape, such as N501Y, E484A, and L452R. The latest research results indicate that the Omicron variant can escape or reduce the neutralization of almost all currently approved vaccines and promising vaccines under development, accompanied by increased morbidity and mortality; therefore, Omicron has attracted widespread attention [84].

The emerging SARS-CoV-2 variants mainly escape the humoral immune response, and some S-protein mutations may also escape the cellular immune response. However, for humoral immunity, antibodies, as the main effectors, can only recognize viral particles that are free in blood and tissue fluid by binding to viral surface proteins, such as S, and cannot recognize viral internal proteins and viral particles that enter cells, which results in a space limitation and rare availability of targets. Therefore, S-protein mutations on the surface of the virus reduce the protective efficacy of the current vaccines that mainly induce humoral immunity and the titers of neutralizing antibodies. Different from humoral immunity, cellular immunity relies on the T-cell receptors (TCRs) on the surface of T cells to recognize the antigen sequences presented by major histocompatibility complex (MHC) molecules on antigen-presenting cells or target cells. Therefore, cellular immunity allows both surface proteins and internal proteins of the virus to be recognized and facilitates the lysis of cells invaded by the virus. Compared with humoral immunity, cellular immunity can recognize a variety of antigens and can attack cells invaded by the virus. Currently, major S-protein mutations are far less likely to evade vaccine-induced cellular immunity than they are to evade humoral immunity. A study evaluated the changes in the cellular immune response to multiple VOCs in COVID-19 convalescents and mRNA-vaccinated people and found that only 3–7% of previously identified wild-type SARS-CoV-2 S-protein epitopes were affected by the mutant strains [85]. The evaluation of VOCs in Ad26.COV2.S-vaccine-immunized population also showed similar results [86]. Further studies evaluated the response changes in the humoral and cellular immunities induced by mRNA, adenovirus, and recombinant subunit vaccines to various strains, including Omicron. It was found that more than 80% of the epitope recognized by cellular immunity was retained even with Omicron compared with the sharply reduced humoral immunity [87]. Based on these results, a recent clinical trial was conducted on a peptide vaccine using epitopes of SARS-CoV-2 proteins recognized by cellular immunity to analyze the ability of the existing variants to escape from this vaccine. The results indicated that this peptide vaccine could induce strong cellular immunity and that the current major VOCs could not escape the cellular immune response induced by the vaccine [88].

### 1.5. Role of Cellular Immunity in the Prevention of Severe Illness Caused by SARS-CoV-2 Variants

The Delta and Omicron variants have higher transmission efficiencies than the original SARS-CoV-2 strain. Many studies have shown that the neutralizing abilities of the convalescent serum of COVID-19 patients, the serum of vaccinated individuals, and therapeutic monoclonal antibodies against Omicron are significantly reduced. Recently, in Israel and the United States, people who have been vaccinated with three doses and those who have recovered from COVID-19 can still be infected by Omicron, leading to a record number of new infections worldwide and further aggravating the prevention and control of the COVID-19 pandemic, and social and economic pressures. Omicron has forced a reconsideration of the effectiveness of vaccines. However, vaccination, whether with an mRNA vaccine, DNA vaccine, adenovirus vaccine, or recombinant subunit vaccine, still provides protective effects against severe disease caused by the Delta and Omicron variants [13]. The goal of SARS-CoV-2 vaccines has been modified from the prevention of infection to the prevention of severe disease caused by SARS-CoV-2 infection. During this process, we have found that the humoral immunity represented by serum neutralizing antibodies seems to lose efficacy, with emerging evidence stemming from the significantly reduced neutralizing capacity of therapeutic monoclonal antibodies against Omicron. Some clinical trials showed that high doses of SARS-CoV-2 therapeutic monoclonal antibodies administration had fewer effects on COVID-19 treatment outcomes [89,90]. In contrast to individuals who seroconverted on their own, the viral loads in subjects treated with therapeutic monoclonal antibodies dropped more slowly even if the antibody injections exhibited 100-fold higher levels of neutralizing antibody titers than native neutralizing antibody responses [90]. Moreover, neutralizing-monoclonal-antibody-treated COVID-19 patients did not achieve improved clinical outcomes either [91]. Therapeutic monoclonal antibodies can generally represent the effect of humoral immunity, and the ineffectiveness of therapeutic monoclonal antibodies against Omicron may partially indicate that humoral immunity has little effect on patients with severe disease. From this perspective, the preventive effect of vaccines against severe disease mainly depends on cellular immune functions. The results are consistent with a dominant role of T cells in the control and clearance of ongoing SARS-CoV-2 infection. 

### 1.6. Cellular Immunity May Have the Potential to Ease the Sequelae of COVID-19

The sequelae of COVID-19 are persistent health problems in the rehabilitated person 4 weeks or 30 days after the onset of the initial symptoms [92,93,94]. Nowadays, the sequelae of COVID-19 have been investigated and reported by several studies. After analyzing the brain scan data of hundreds of COVID-19 patients, one group found that the thickness of gray matter in the orbitofrontal cortex and para-hippocampal gyrus of the infected patients was significantly thinner than that of the uninfected ones 5 months after the infection with SARS-CoV-2 [95]. Similarly, another team analyzed the clinical data of more than 3000 patients aged 60 and over and their family members from February to April 2020 and evaluated the cognitive changes in these patients one year after rehabilitation. They found that the overall incidence of cognitive impairment in infected patients was as high as 12.45%. The incidences of dementia and mild cognitive impairment (MCI) in severe patients were 15% and 26.54%, respectively. In contrast, the incidence of dementia was less than 1%, and the incidence of MCI was around 5% in mild patients, which further confirmed the negative effects of SARS-CoV-2 on the brain [96]. Besides the long-term impact of COVID-19 on the nervous system, studies have counted more than 50 long-term effects of COVID-19 on the human body and found that more than 80% of people infected with SARS-CoV-2 are likely to suffer of at least one sequela [97]. The risk of sequelae of COVID-19 is much higher than that of seasonal influenza [98]. The five most common types are fatigue (58%), headache (44%), attention disorder (27%), hair loss (25%), and dyspnea (24%). Other symptoms include lung disease (cough, chest discomfort, pulmonary diffusion capacity, sleep apnea, and pulmonary fibrosis), cardiovascular disease (arrhythmia, myocarditis), neurological and psychiatric disorders (dementia, depression, anxiety, obsessive compulsive disorder), etc. [99,100,101,102]. 

A study found multiple early risk factors for the occurrence of sequelae or long-term COVID-19. Female sex, obesity, and invasive mechanical ventilation were identified as the factors associated with being less likely to report full recovery after 1 year [103]. Researchers also found that reactivated Epstein-Barr virus after SARS-CoV-2 infection, the presence of specific autoantibodies related to lupus and other autoimmune diseases, the viral load in the blood in the early stage of SARS-CoV-2 infection, and type II diabetes are closely related to the long-term COVID-19 symptoms after rehabilitation [104,105].

From the above, it is easy to see that the influence of COVID-19 on the human body almost involves all tissues or organs for the reason that the human angiotensin converting enzyme 2 (ACE2) mediating SARS-CoV-2 entering into cells is widely expressed by various tissues or organs.

ACE2-mediated SARS-CoV-2 widespread distribution is also one of the most likely mechanisms to induce sequelae. When the symptoms of pneumonia disappear and the throat swab is negative, the virus may still reproduce in other target organs. For example, one study detected viral presence in multiple organs involving the heart, brain, liver, kidneys, and blood of COVID-19 patients. Another detection of 14 negative throat swabs of asymptomatic patient’s intestinal tissue also found SARS-CoV-2 [106,107]. Therefore, the first possible mechanism is the direct cell or tissue damage caused by SARS-CoV-2 infection. Studies have shown that SARS-CoV-2 replication in endothelial cells causes serious damage to the lungs and respiratory tract in the respiratory system, which is similar to the cardiovascular system [108,109]. In other tissues and organs, the infection of SARS-CoV-2 can also lead to the death of related cells, thereby affecting the function of organs and inducing the sequelae of COVID-19. The potential viral library or non-infectious SARS-CoV-2 fragment can also trigger chronic inflammation and cause tissue damage. It may be due to the persistence of virus fragments, leading to chronic inflammation and causing the sequelae of COVID-19 [110]. The second possible cause is abnormal immune metabolism and mitochondrial dysfunction. Mitochondria are not only the energy factories of cells but are also crucial to the immune homeostasis of the human body. Therefore, damage to mitochondrial functions inevitably affects human immunity. As early as the outbreak of SARS-CoV-2, scientists have studied the interaction between the protein encoding by SARS-CoV-2 and human mitochondria. Based on these studies, scientists have speculated that SARS-CoV-2’s non-structural proteins (NSPs) 4 and 8, and ORF9c could interact with mitochondria. Studies have also found that SARS-CoV-2 can hijack host mitochondria [111]. From clinical research data, the peripheral blood mononuclear cells (PBMCs) of patients with SARS-CoV-2 showed the characteristics of mitochondrial dysfunction, metabolic changes, and high levels of mitochondrial factors [112]. On the one hand, mitochondrial dysfunction leads to the imbalance of immune homeostasis, and on the other hand, it also leads to the metabolic reprogramming of infected cells. The two may be behind the sequelae of COVID-19.

The third guess is that COVID-19 is associated with immune exhaustion. As we all know, if immune cells are stimulated by antigens for a long time, they present with dysfunction or even exhaustion. Immune exhaustion is a phenomenon associated with chronic viral infection. The main feature of immune exhaustion is the functional disorder of antigen-specific immune cells caused by long-term antigen stimulation, including reduced cytokine production, impaired clonal proliferation, and up-regulated expressions of inhibitory receptors [113,114]. It was found that the absolute numbers of antiviral lymphocytes such as cytotoxic T lymphocytes (CTLs) and natural killer cells (NKs) were significantly reduced and their functions were impaired in patients with severe COVID-19 [115]. Other studies found that the expression levels of many immunosuppressive receptors on lymph and bone marrow cells were up-regulated during SARS-CoV-2 infection. This immunosuppression and depletion of immune cells may promote SARS-CoV-2 infection and lead to COVID-19 sequelae [102] (Figure 2).

An immune response analysis of COVID-19 patients with or without sequelae showed that the level of S-specific CD8^+^ T cells decreased faster than that in patients without sequelae, indicating that a weakened cellular immune response was associated with sequelae [115].

Therefore, the rapid elimination of SARS-CoV-2 from tissues through vaccine-induced pre-existing cellular immunity may reduce the occurrence of sequelae of COVID-19. On the topic of whether vaccination is associated with a decreased prevalence of post-acute sequelae of SARS-CoV-2 infection compared with no vaccination, two recent studies on the association between vaccination and COVID-19 sequelae reached opposite conclusions [116,117]. However, for patients with sequelae, vaccine immunization could still arouse a strong cellular immune response. On the one hand, this study showed that the cellular immune response did not enhance the sequelae. On the other hand, it also showed that the patients with sequelae may have virus antigens all the time and the reaction could be strong [118]. However, due to the difficulty of evaluating sequelae, the relationship between vaccination-induced cellular immunity and sequelae has not yet been determined, and further research is needed. Further observation and comparison of the incidence of sequelae between patients with genetic and drug-induced B-cell loss may contribute to further understanding the role of cellular immune responses in controlling sequelae.

### 1.7. Strategy for Cellular Immunity Vaccine Design and Methods for Cellular Immunity Evaluation

For vaccine design, the first thing to consider is the choice of antigen. The target used to design a cellular immune vaccine cannot only contain the S protein, but membrane (M), nucleocapsid (N), and non-structural proteins are also important options. Bioinformatics analysis and experiments have proved that these proteins contain epitopes that can be recognized by T cells [119,120]. Evidence demonstrating the efficacy of the vaccines with these non-S proteins has increased. Accordingly, the viral load in the lungs of the RBD mRNA-vaccine-immunized group was equivalent to that of the non-vaccinated group. However, the viral load in the lungs of the N-protein mRNA-vaccine-immunized group significantly decreased, which indicated that an mRNA vaccine containing N antigens could reduce the viral load in the lungs and avoid the weight loss of mice [77]. Adjuvants are still essential for vaccines to reinforce the cellular immune response. Adjuvants need to be properly selected according to the characteristics of different vaccine platforms. Various adjuvants applicable to conventional vaccine platforms and the possible underlying mechanisms to enhance the immune response have been fully summarized elsewhere [75]. Molecular adjuvants could be adopted for nucleic acid vaccines. Helping to enhance the cellular immune response is a new strategy for vaccine design. Molecular adjuvants can be encoded and expressed simultaneously with antigens. Many studies have confirmed that the addition of molecular adjuvants can significantly enhance the intensity of the antigen-induced immune response. For instance, the induction of more potent CD8^+^ T-cell cytotoxicity was obtained by the fusion of antigens with XCL1 [121]. After immunization, the most commonly used method to evaluate specific T-cell immunity is to re-stimulate T cells with the peptide pool covering the antigen. Cytokines and markers of different functional subgroups of T cells detected by flow cytometry can be used to define the type of activated cellular immune response and analyze the induced specific T-cell subsets. For example, circulating follicular helper T (cTfh) cells can be characterized by CD4^+^, OX40^+^, surface CD40L^+^, and CXCR5^+^ [13]. If the CD8^+^ T-cell recognition epitope sequence can be identified, then the MHC I molecular tetramer could be prepared to detect specific CD8^+^ T cells. Furthermore, the combination of KLRG, CD127, and CD62L can distinguish specific CD8^+^ T cells into central-memory or effect-memory CD8^+^ T cells [122].

## 2. Conclusions

Recently, two small-molecule drugs have significantly improved the symptoms of patients with COVID-19 by reducing the viral load in the body by inhibiting viral replication [123,124]. For the immune system, the way to control the viral load is to lyse virus-infected cells through cellular immunity, suggesting the necessity of designing vaccines with a strong cellular immune response for the reasons that cellular immunity can reduce the severity of the disease, is still effective against the mutant strain, and can prevent the sequelae. The virus particles inside virus-invaded cells that are released after the cells are lysed by cellular immune responses still require antibodies for clearance. Therefore, the orchestration of humoral immunity and cellular immunity can maximize immune protection. However, it is difficult for current vaccines to strongly induce both cellular and humoral immune responses. With the emergence of variants, booster vaccinations have been implemented in many countries and regions. Therefore, to achieve better protection, whether the boosters should be the original vaccines or other vaccines with complementary immune responses has become an issue that needs to be explored. Moreover, the combined immunization with vaccines that induce different subtypes of cellular immune responses to generate more comprehensive cellular immunity is also a scheme worthy of consideration. In addition, the current nucleic acid vaccines do not contain molecular adjuvants, which can help to adjust the type of induced immunity. Adding appropriate molecular adjuvants to help to enhance the cellular immune response is a new strategy for vaccine design. In summary, the SARS-CoV-2 pandemic provides a broad application scenario for vaccines based on various platforms and provides a real-world test platform for many previously unevaluable immunization strategies. This unprecedented public health crisis requires bold attempts from humankind. 

## Figures and Tables

**Figure 1 vaccines-10-01103-f001:**
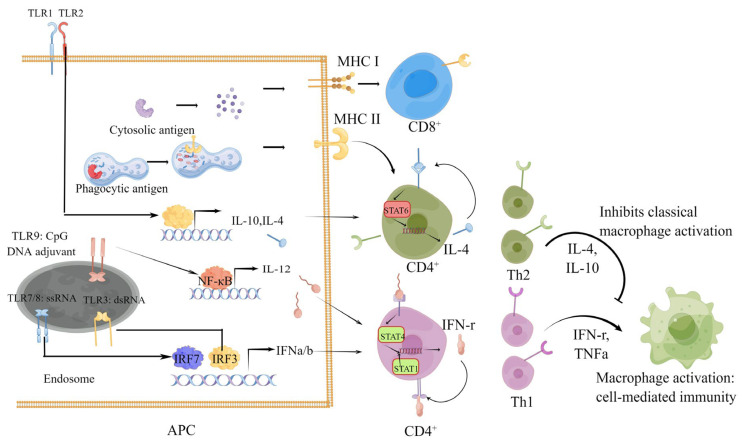
The underlying mechanism of vaccine antigen delivery forms and adjuvants shape the type of cellular immune response. Generally, the antigens of the conventional vaccine platform are degraded in lysosomes and then subjected to MHC II to be presented to CD4^+^ T cells after being phagocytized by APCs. The antigens of nucleic acid vaccines that are expressed and degraded in the cytoplasm of APCs interact with MHC I molecules to be presented to CD8^+^ T cells, while the antigens expressed in other non-APCs is similar to the conventional vaccine platform. The adjuvant that stimulates the TLR1/-2 on the cell membrane promotes APCs secreting IL-4 and IL-10 and help Th2 differentiation. TLR3, -7/-8, and -9 elevate the expression of IL-12 and IFN a/b and induce Th1 differentiation through different transcription factors upon activation.

**Figure 2 vaccines-10-01103-f002:**
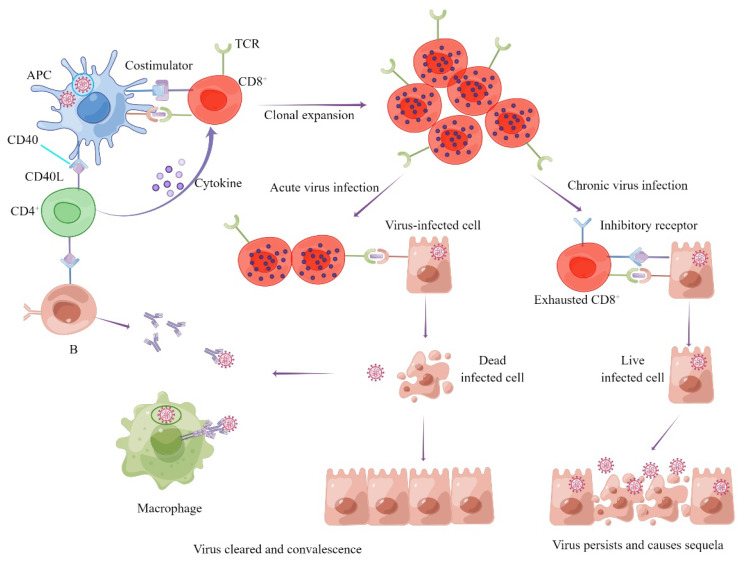
The underlying mechanisms of clearing SARS-CoV-2 or virus latency and causing sequelae.

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
