# Peer review of "The Role of Cellular Immunity in the Protective Efficacy of the SARS-CoV-2 Vaccines"

_vaccines, 2022, doi:10.3390/vaccines10071103_

Round 1

Reviewer 1 Report

The manuscript discussed the roles of cellular immune responses in the protective efficacy of vaccines. I think it is important not to ignore cellular immunity when evaluating the efficacy of vaccines. The authors cited many references to emphasize that neutralizing antibodies may not the only one factor associated with severity.

We expect to see more paper studying the protective effects of cellular immunity on variants and COVID sequelae.

I have few comments of the manuscript:

1.      In Table 1, in the column of “Clinical efficacy”, there were two “No publication”.“Not mentioned” may be more appropriate.

2.      In the abstract, Line 35-37 should be re-written.

3.      In Fig.1, I suggest more explanation should be given since it tried to explain the mechanisms how vaccine delivery and adjuvant shape the immune responses.

4.      I would expect to see a paragraph in the text describing the vaccine design and immunization strategy to elicit cellular immunity since it was mentioned in the last sentence of abstract.

Reviewer 2 Report

This review describes the role of cellular immunity in the protective efficacy of the SARS-CoV-2 vaccines. This is an interesting review; the authors describe in detail the role of cellular immunity against COVID-19 and summarize recent advances in the development of SARS-CoV-2 vaccines and the immune responses induced by vaccines to raise the new vaccine design and immunization strategy. I have a few minor suggestions to improve the quality of the article.

Minor comments

1.      In the introduction (Line 77), the authors mentioned that the preventive efficacy of different vaccines is significantly variable. Instead of giving a vague the authors should specify the range of efficacy of different vaccines.

2.      In line 78, the authors describe neutralizing antibodies? The authors should briefly discuss the formation of neutralizing antibodies like duration of production, type of antibodies the half-life of the antibodies.

3.      The authors should also incorporate the details of the method of estimation of cellular immunity induced by vaccines in coronavirus vaccines.

4.      Authors discuss the long-term impact of the COVID-19 on the nervous system, studies have counted more long-term effects of the COVID-19 on the human body, and found that more than 80% of people infected with the SARS-CoV-2 will suffer at least one sequelae? What are the predisposing factors for the development of these sequels? The authors can be specific on these?

5.      There are a few minor typo errors, authors can use any free software to avoid these.
